# Echocardiographic Assessment of Pulmonary Hypertension in Neonates with Congenital Diaphragmatic Hernia Using Pulmonary Artery Flow Characteristics

**DOI:** 10.3390/jcm11113038

**Published:** 2022-05-27

**Authors:** Florian Kipfmueller, Suemeyra Akkas, Flaminia Pugnaloni, Bartolomeo Bo, Lotte Lemloh, Lukas Schroeder, Ulrich Gembruch, Annegret Geipel, Christoph Berg, Andreas Heydweiller, Andreas Mueller

**Affiliations:** 1Department of Neonatology and Pediatric Intensive Care, Children’s Hospital, University of Bonn, 53113 Bonn, Germany; suemeyra.akkas@web.de (S.A.); flaminia.pugnaloni@gmail.com (F.P.); bartolomeo.bo@uni-bonn.de (B.B.); lotte.lemloh@gmx.de (L.L.); lukas.schroeder@ukbonn.de (L.S.); a.mueller@ukbonn.de (A.M.); 2Center for Rare Diseases Bonn, Division of Congenital Malformations, University of Bonn, 53113 Bonn, Germany; andreas.heydweiller@ukbonn.de; 3Department of Neonatology, Ospidale Bambinu Gesu, 00165 Rome, Italy; 4Department of Obstetrics and Prenatal Medicine, University Hospital Bonn, 53127 Bonn, Germany; ulrich.gembruch@ukbonn.de (U.G.); annegret.geipel@ukbonn.de (A.G.); christoph.berg@ukbonn.de (C.B.); 5Division of Prenatal Medicine, University Hospital Cologne, 50931 Cologne, Germany; 6Division of Pediatric Surgery, Department of General, Visceral, Thoracic and Vascular Surgery, University of Bonn, 53113 Bonn, Germany

**Keywords:** congenital diaphragmatic hernia, echocardiography, extracorporeal membrane oxygenation, pulmonary artery acceleration time, pulmonary hypertension

## Abstract

Background: Assessment of pulmonary hypertension (PH) is essential in neonates with congenital diaphragmatic hernia (CDH). Echocardiography is widely established to quantify PH severity, but currently used parameters have inherent limitations. The aim of our study was to investigate the prognostic utility of the index of the pulmonary artery acceleration time to the right ventricular ejection time (PAAT:ET) in CDH neonates assessed using echocardiography. Methods: PAAT:ET values were prospectively measured in CDH neonates on admission, on day of life (DOL) 2 and DOL 5–7. Optimal cut-off values to predict mortality and need for ECMO were calculated and PAAT:ET values were compared between non-ECMO survivors, ECMO-survivors, and ECMO-non-survivors. Results: 87 CDH neonates were enrolled and 39 patients required ECMO therapy. At baseline, PAAT:ET values were significantly lower in ECMO patients compared to non-ECMO patients (*p* < 0.001). ECMO survivors and ECMO non-survivors had similar values at baseline (*p* = 0.967) and DOL 2 (*p* = 0.124) but significantly higher values at DOL 5–7 (*p* = 0.003). Optimal PAAT:ET cut-off for predicting ECMO was 0.290 at baseline and 0.310 for predicting non-survival in patients on ECMO at DOL 5–7. Conclusion: PAAT:ET is a feasible parameter for early risk assessment in CDH neonates.

## 1. Introduction

Congenital diaphragmatic hernia (CDH) occurs in approximately 1 in 2500 pregnancies and continues to be associated with high mortality [1]. Pulmonary hypertension (PH), lung hypoplasia and cardiac dysfunction are major contributors to morbidity and mortality among affected infants [2,3,4,5]. According to recent registry-based data, approximately 85% CDH newborns present with PH within the first 48 h of life with limited treatment options [6,7]. Echocardiography is the most practical bedside technique to assess PH severity [4,8]. The classification of PH severity includes qualitative and quantitative measures, such as the flow direction in the patent ductus arteriosus (PDA), the tricuspid regurgitation (TR) velocity, and the position of the intraventricular septum (IVS). However, each of these echocardiographically determined parameters have limitations, relating to their prognostic value or interobserver variability. CDH newborns receiving extracorporeal membrane oxygenation (ECMO) support represent the subgroup with the highest mortality (40–50%), compared to survival approaching 100% in cases who do not fulfil ECMO criteria [9]. This underlines the importance of early and repeated risk stratification of CDH neonates during ECMO support.

Echocardiographic measurements of the pulmonary artery acceleration time (PAAT; also known as “time to peak velocity”), and the relation of the PAAT to the right ventricular ejection time (ET) are increasingly used to assess the pulmonary artery pressure (PAP) in infants, children and adults with PH [10,11,12,13]. In this context, a shorter PAAT and a lower PAAT:ET ratio are indicative of increased pulmonary artery pressure (PAP) [12,14]. The prognostic role of PAAT:ET values in CDH neonates is not well investigated [15,16,17]. In a retrospective study, a lower PAAT:ET (≤0.290) measured within the first hours of life was associated with an increased need for ECMO support [15]. The aim of this prospective, observational study was to investigate the association of repeated PAAT:ET measurements with outcome parameters in CDH neonates.

## 2. Methods

### 2.1. Study Design

CDH newborns treated in our institution between April 2015 and April 2019 were eligible for study enrollment. Inclusion criteria were diagnosis of CDH and an echocardiography study performed within 12 h of admission. Exclusion criteria were non-isolated CDH, palliative care, contraindication to ECMO therapy, and late-presenting CDH. The Institutional Review Board approved the study. Informed consent was obtained from parents or legal representatives prior to enrollment in the study.

### 2.2. Treatment Protocol

Infants were intubated after delivery and mechanical ventilation was started using a maximum peak inspiratory pressure of 20–26 cmH_2_O, a positive end-expiratory pressure of 3–5 cmH_2_O and a respiratory rate of 60 to 80/min. Initial inspired oxygen fraction (FiO_2_) was 1.0 and infants received iNO therapy during initial period of stabilization. FiO_2_ was titrated to achieve a postductal partial oxygen pressure (PaO_2_) of 80–150 mmHg and ventilator settings were adjusted to reach a pCO_2_ of 45–60 mmHg. Dobutamine or milrinone was administered to treat cardiac dysfunction, and norepinephrine and vasopressin were added to achieve a mean arterial blood pressure of ≥40 mmHg. Infants were sedated with fentanyl and midazolam during the first days of life. Criteria for ECMO were according to published guidelines of the CDH-Euro Consortium [18]: preductal oxygen saturation < 85% or postductal saturation < 70%, OI ≥ 40 consistently present, increased PaCO_2_ > 70 mmHg with a pH < 7.15, a peak inspiratory pressure ≥ 28 cmH_2_O or mean airway pressure ≥ 17 cmH_2_O, or persistent systemic hypotension (mean arterial pressure < 40 mmHg). ECMO support was performed with the Deltastream DP3 system (Xenios, Aachen, Germany), a rotational pump with a diagonally streamed impeller [19]. The preferred mode was veno-venous (v-v) ECMO, although veno-arterial (v-a) ECMO was applied in infants with vessels being too small for a 13-French cannula. Surgical repair was performed after an initial period of stabilization and in ECMO patients after weaning from ECMO (delayed repair).

### 2.3. Echocardiography Data

Echocardiography was obtained using a Philips CX50 Compact Extreme Ultrasound System with an S12-4 sector array transducer (Philips Healthcare, Best, The Netherlands). The routine use of echocardiography for clinical decision making was permitted at the discretion of the attending neonatologist. For study purposes echocardiography was performed after admission (baseline), at the second day of life (DOL 2) in ECMO patients approximately 24 h after starting ECMO, and at 5–7 days of life (DOL 5–7).

The pulmonary arterial pressure (PAP) was graded as <⅔ systemic pressure (mild PH), ⅔ to systemic pressure (moderate PH), or suprasystemic pressure (severe PH) as described by Keller et al. [3]. Assessment of PH included: (1) DA flow pattern; (2) intraventricular septum position; and (3) calculation of right ventricular systolic pressure from the TR jet with an estimation of 5mmHg for right atrial pressure. The angle of insonation was kept below 20°. To obtain PAAT:ET measurements, the main pulmonary artery (PA) was identified in the parasternal short axis view or from the subcostal window and the echo transducer was placed directly distal to the pulmonary valve. PAAT was measured by PW Doppler from the beginning of systolic flow in the main PA to the point of the highest velocity during systole, while the ET was measured as the duration of the systolic blood flow (Figure 1). Using the incorporated measurement calipers, both values were expressed in milliseconds (ms).

### 2.4. Physiologic, Treatment and Outcome Data

Information on treatment and outcome data were prospectively documented at the time of the echocardiography study. The primary clinical endpoint was defined as mortality. The secondary clinical endpoint was need for ECMO, or early mortality in cases with contraindications to ECMO despite fulfilling ECMO criteria.

### 2.5. Statistical Analysis

SPSS Version 26 (IBM Corp. Armonk, NY, USA) was used for statistical analysis. Data were described as median (interquartile range) or absolute number (percentage). For the statistical analysis patients were allocated according to their outcome to one of three groups: non-ECMO survivors, ECMO survivors, and ECMO non-survivors. The distribution of baseline characteristics and PAAT:ET values between groups were compared using the Mann–Whitney-U-test or the Kruskal–Wallis test as appropriate. Using receiver operating characteristic (ROC) analyses, the optimal PAAT:ET cutoff values for predicting the primary and secondary clinical endpoints at baseline, DOL 2 and DOL 5–7 were calculated. Additionally, PAAT:ET values were compared in ECMO survivors and ECMO non-survivors. Sensitivity, specificity, negative and positive predictive values for the respective cutoff values were determined. Spearman’s rank or Pearson’s coefficient were used to determine correlations between PAAT:ET and PH severity, gestational age, and defect size. The Kaplan–Meier estimator and log-rank test were used to estimate the cumulative probability of the clinical endpoint in neonates with a PAAT:ET above and below the respective cutoff values. A *p*-value of <0.05 was considered to indicate statistical significance.

## 3. Results

During the study period, 109 CDH newborns were treated at our institution. However, 22 patients were not included in the study for the following reasons: palliative care (n = 2); major congenital heart defect (n = 5); late presenting CDH (n = 6); families not approached by study personnel (n = 7); or start of ECMO support post-surgical repair at DOL 6 and DOL 7, respectively. Therefore, the final cohort consisted of 87 prospectively enrolled CDH neonates. The baseline characteristics are presented in Table 1. Overall, survival to discharge was 80.5% and the ECMO rate was 44.8% (n = 39). Three patients received ECMO for <5 days and the PAAT:ET values of these patients were not included in the analysis for DOL 5–7.

PH severity on echocardiography during the first week of life in ECMO and non-ECMO patients is demonstrated in Figure 2.

At baseline, 37.9% and 40.2% presented with moderate and severe PH, respectively. During echo assessment at baseline, DOL2 and DOL 5–7, a PDA was visible in 97%, 72%, and 32% of patients and a TI in 62%, 61%, and 30%. The IVS and the pulmonary artery were appropriately visible for assessment at all timepoints. PAAT:ET values according to group allocation to non-ECMO survivors, ECMO survivors, and ECMO non-survivors are shown in Figure 3.

Non-survivors had a significantly lower median PAAT:ET compared to survivors at baseline, DOL 2 and DOL 5–7. Additionally, the median PAAT:ET at baseline was significantly lower in patients subsequently receiving ECMO support compared to non-ECMO-patients (*p* < 0.001). Comparing survivors and non-survivors among the 39 ECMO patients, the median PAAT:ET values were similar at baseline and DOL 2, but significantly lower in non-survivors at DOL 5–7 (Figure 3). The median PAAT:ET value at DOL 2 and DOL 5–7 was 101% and 103% of the baseline value in ECMO non-survivors and 128% and 139% in ECMO survivors, respectively. Further, 35.3% and 31.3% of ECMO non-survivors and 54.5% and 72.2% of ECMO survivors presented an increase of more than 10% from baseline to DOL 2 and DOL 5–7, respectively. The ECMO rate was 79.5% in patients with a PAAT:ET ≤0.290 at baseline and 17.4% with a PAAT:ET above 0.290 (*p* < 0.001). Using a cutoff of 0.256 at baseline, the mortality rate was 48.0% and 8.1% in patients with a PAAT:ET below and above the cutoff, respectively. For each timepoint of echo assessment, the area under the curve (AUCs), optimal cutoff values, number of patients ≤ cutoff point, sensitivity, specificity, positive predictive value, negative predictive value, and the relative risk at the optimal cutoff point for predicting both mortality and ECMO are presented in Table 2.

Kaplan–Meier survival curves for the respective PAAT:ET cutoff values are provided in Figure 4. The cumulative mortality rate through DOL 150 was significantly higher in patients with a PAAT:ET ≤0.256 at baseline (Figure 4a, log rank, *p* < 0.001). On DOL 5–7, neonates on ECMO with a PAAT:ET ≤0.310 had a significantly higher cumulative probability for survival (Figure 4b, log rank, *p* = 0.012).

PAAT:ET values at baseline, DOL 2, and DOL 5–7 demonstrated a significant negative correlation with PH severity at baseline (r = −0.492; *p* < 0.001), DOL 2 (r = −0.730; *p* < 0.001), and DOL 5–7 (r = −0.511; *p* < 0.001), respectively. There was a significant correlation of defect size with PAAT:ET at baseline (r = −0.308; *p* = 0.004), DOL 2 (r = −0.335; *p* = 0.001), and DOL 5–7 (r = −0.504; *p* < 0.001). PAAT:ET values did not correlate with gestational age. The mortality rate in association with PH severity at baseline was 0% for mild/no PH (n = 19), 9.1% for moderate PH (n = 33), and 40% for severe PH (n = 35).

## 4. Discussion

PH, lung hypoplasia, and cardiac dysfunction can be considered as the cornerstones of morbidity and mortality in CDH neonates [7,20]. Several studies have demonstrated the association between PH severity, especially suprasystemic PH, and poor outcome [7]. However, there is no unified definition for PH in the early neonatal period and each method of assessment has specific limitations [21]. In the current study, we evaluated the prognostic information derived from PAAT:ET measurements as a marker of PVR and PH in CDH neonates. We observed significantly lower PAAT:ET values in the first week of life (i.e., after admission, on DOL 2, and DOL 5–7) in non-survivors compared to survivors. In our cohort, almost one in three patients had a PAAT:ET ≤ 0.256 at admission, and in those, the risk of death increased approximately 6-fold. Conversely, patients with a PAAT:ET >0.256 had a survival rate of 91.9%. Additionally, neonates with need for early ECMO support (i.e., before surgical repair) had significantly worse PAAT:ET values and patients with a PAAT:ET ≤0.290 at admission demonstrated a four-times higher early ECMO rate. In the subgroup of patients receiving ECMO support, PAAT:ET values were significantly worse in non-survivors compared to survivors during ECMO on DOL 5–7, but not within the first 24 h after ECMO cannulation (DOL 2). Therefore, using the PAAT:ET seems reasonable for predicting both the requirement of ECMO and the response to ECMO treatment.

Based on earlier findings, this study was designed to prospectively investigate the feasibility of PAAT:ET measurements for the prediction of outcome in CDH neonates. Our primary aim was to assess the risk of mortality and the need for ECMO in this population. The three timepoints for echocardiographic assessment within the first week of life were chosen based on a presumed relevance for risk assessment in CDH neonates [22,23]. Echo assessment should be performed early after birth and repetitively thereafter [22]. In our NICU, echocardiography is frequently used in patients during ECMO support; however, for study purposes, the timing of echo assessment was set at 24 h after ECMO initiation and at 5 days on ECMO. In non-ECMO patients, these dates were adapted, given the fact that ECMO initiation is usually performed within the first 24–48 h of life.

Echocardiography is a widely established bedside tool to assess PH in CDH neonates. Several echo parameters exist to evaluate PH severity in these patients. In CDH neonates, most commonly, a grading system of PH severity introduced by Keller and colleagues based on the TR jet velocity, the PDA flow, and the IVS position is used [3,4]. Although this is a straightforward approach, both the PDA and the TR may not be assessable in a relevant proportion of neonates after birth or during the course of the disease [4]. Therefore, there is a need for a thorough evaluation of additional parameters. This is supported by previously published data by Lusk et al. and the findings from our study, showing that the proportion of CDH newborns with either a PDA or a TR is decreasing from 98% at baseline to 43% on DOL 5–7. Although CDH neonates on ECMO had a higher proportion, this did not exceed 67% on DOL 5–7. Additional potential parameters include the foramen ovale flow pattern, a pulmonary valve insufficiency jet, the tricuspid annular plane systolic excursion (TAPSE), and the left ventricular eccentricity index [11,24,25]. The limitations with these parameters are reduced interobserver reproducibility, inconsistent availability, or an insufficient evaluation in CDH neonates. In our study, the main PA was visible to obtain PAAT:ET measurements in all included patients and was, therefore, more consistently available than a TR jet or a PDA flow pattern. This is in agreement with results from previous studies reporting PAAT and PAAT:ET measurements being feasible in >95% of patients, with high interobserver reliability [11,12]. PAAT:ET measurements are relatively easy to obtain in most neonates with high reproducibility, although results might differ slightly when measured proximally or distally to the pulmonary valve [26].

Levy and colleagues demonstrated a significant correlation of PAAT and PAAT:ET measurements with the invasively determined mean PAP, PVR, and compliance using right heart catheterization [27]. However, in their study, the median age at study participation was 5.3 years and studies comparing invasive and non-invasive methods have not been conducted in neonates [27]. A recent study in a cohort of infants (i.e., <1 year of age) referred for cardiac catheterization demonstrated only a weak correlation of invasively measured PAP and PAAT values using echocardiography in patients with a PDA [28]. However, this study is limited by a small sample size and differences in the sedation protocol during echocardiography and heart catheterization [28]. Theoretically, PAAT:ET measurements might have a lower predictive value in patients with a large right-to-left shunt via the PDA. In this setting, the PDA might function as a blow-off, leading to a pressure drop in the main PA [22]. Presumably, PAAT:ET values might not only be affected by the shunt direction but also by the size of the PDA. Based on our findings this might play a minor role during the first week of life, but we did not prospectively quantify PDA flow or PDA size. Notably, prostaglandin E1 was not used in this study cohort to maintain ductal patency.

PAAT:ET measurements have been used for risk stratification in CDH neonates in previous studies [15,16,17]. Baptista et al. reported significantly higher PAAT:ET values in survivors compared to non-survivors at 24 h of age. In a retrospective study, including 40 CDH neonates, a PAAT:ET ≤ 0.290 was associated with a 5.9-fold risk of ECMO [15]. Recently, Aggarwal et al. demonstrated a similar pattern of significantly better PAAT and PAAT:ET values in CDH survivors without ECMO compared to patients who died or received ECMO [16]. Our study is the first prospective study measuring PAAT:ET values in CDH neonates.

The use of pulmonary vasodilators did not differ between groups at the time of the first echocardiography study and was also similar on DOL 2 and DOL 5–7 in ECMO patients, regardless of survival or non-survival. Our treatment approach incorporates the early initiation of pulmonary vasodilators, such as iNO, intravenous sildenafil, milrinone and levosimendan [29,30]. However, this study was not designed to investigate a direct effect of these agents on PAP and PH, and it remains uncertain whether improving PAAT:ET values in surviving patients result from the use of this medication or reflect a physiological decline in the PVR over time.

Other factors potentially affecting PAAT:ET measurements, such as body surface area, age, and gender, do not play a major role in the first week of life and are, therefore, negligible [12]. Future studies should investigate a combined approach of different echocardiographic parameters and commercially available biomarkers, such as proBNP. ProBNP has been demonstrated to be useful for risk assessment in CDH neonates before, during and after ECMO therapy, despite high intraindividual variation in the early postnatal transitional period [31,32,33].

## 5. Conclusions

PAAT:ET measurements provide relevant prognostic information during the first week of life in CDH neonates. Lower PAAT:ET values are associated with a higher need for ECMO and increased mortality. Accordingly, the measurement of PA flow pattern should be included in echocardiographic assessment algorithms for these patients. To account for any effects of the study center treatment approach, these finding should be validated in a multi-center study.

## Figures and Tables

**Figure 1 jcm-11-03038-f001:**
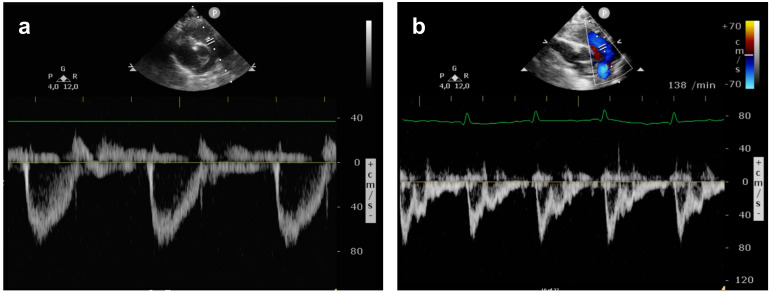
Pulmonary artery flow pattern on the first day of life: (**a**) normal pulmonary artery flow in a healthy neonate showing a PAAT:ET of 0.335; (**b**) pulmonary artery flow in a CDH neonate with severe pulmonary hypertension and a PAAT:ET of 0.146.

**Figure 2 jcm-11-03038-f002:**
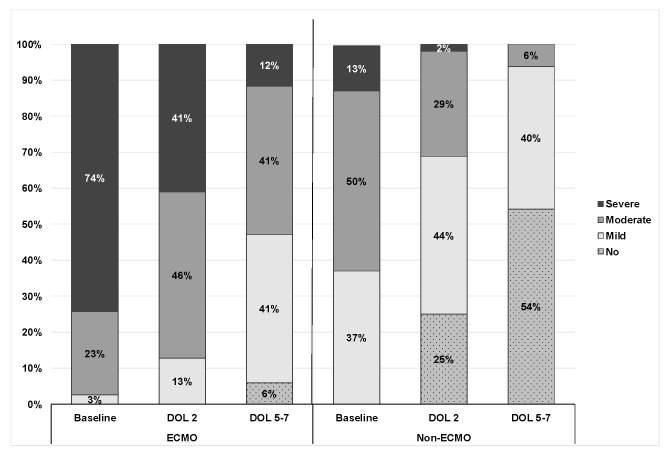
PH severity in the first week of life in ECMO and non-ECMO patients.

**Figure 3 jcm-11-03038-f003:**
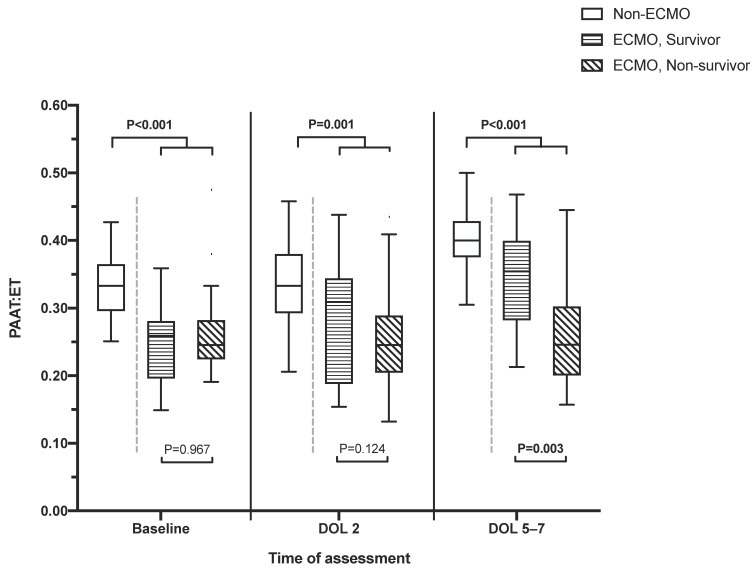
PAAT:ET values at baseline, DOL 2, and DOL 5–7 according to group allocation.

**Figure 4 jcm-11-03038-f004:**
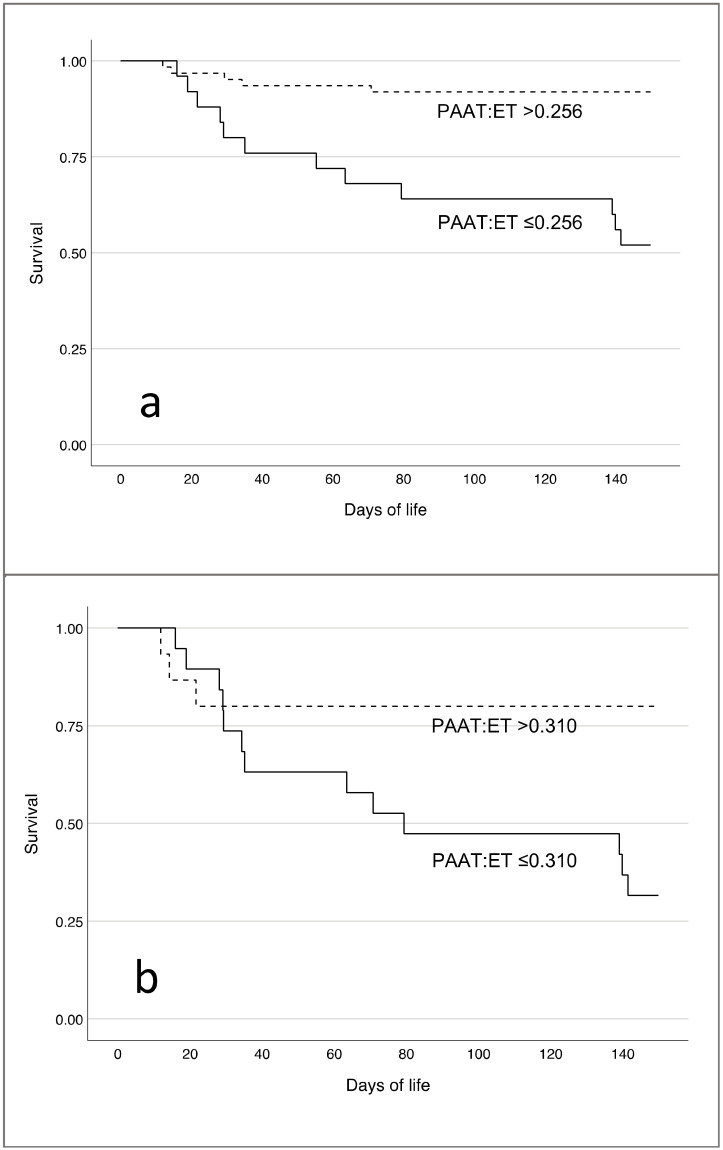
Cumulative survival for patients with a PAAT:ET cutoff of 0.256 at baseline (**a**) and 0.310 on day 5–7 in ECMO patients (**b**).

**Table 1 jcm-11-03038-t001:** Baseline characteristics according to group allocation. *p*-values below 0.05 are highlighted in bold. CDH, congenital diaphragmatic hernia; ECMO, extracorporeal membrane oxygenation; o/e LHR, observed-to-expected lung-to-head-ratio.

	Entire Cohort			ECMO Patients		
Variables	Non-ECMO Group	ECMO Group	*p*-Value	ECMO Survivor	ECMO Non-Survivor	*p*-Value
	n = 48	n = 39		n = 22	n = 17	
Gestational age (days)	38.1 (37.0–39.0)	37.9 (36.0–38.4)	0.138	38.0 (36.6–38.4)	37.6 (35.7–38.3)	0.107
Left-sided CDH, n	44 (91.7%)	29 (74.4%)	**0.030**	81.8%)	64.7%	0.377
O/e LHR, %	44.5 (37.3–53.8)	32.0 (29.0–42.0)	**<0.001**	36.5 (29.5–43)	31 (29–37)	0.179
Liver-up, n	15 (31.3%)	30 (76.9%)	**<0.001**	13 (59.1%)	17 (100%)	**0.029**
Prenatal diagnosis	45 (93.8%)	37 (94.9%)	0.824	20 (90.9%)	17 (100%)	0.644
FETO	4 (8.3%)	9 (23.1%)	0.056	5 (22.7%)	4 (23.5%)	0.967
ECMO						
Survival	48 (100%)	22 (56.4%)	**<0.001**			
Defect Size			**<0.001**			0.063
A	7 (14.6%)	1 (2.6%)		1 (4.5%)	0	
B	24 (50.0%)	3 (7.7%)		2 (9.1%)	1 (5.9%)	
C	15 (31.3%)	15 (38.5%)		10 (45.5%)	5 (29.4%)	
D	2 (4.2%)	16 (41.0%)		9 (40.9%)	7 (41.2%)	
N/R	0	4 (10.3%)		0	4 (23.5%)	
Death on device		5 (12.8%)		0	5 (29.4%)	0.113
Age at ECMO inititiation, hours		9.0 (5.8–21.8)		14 (7.7–23.8)	7.2 (5.0–10.4)	**0.045**
Duration of ECMO, days		8.1 (5.6–16.0)		6.7 (5.8–8.1)	11.8 (9.4–24.8)	**0.001**

**Table 2 jcm-11-03038-t002:** Area under the curve (AUC), sensitivity, specificity, positive predictive value (PPV), negative predictive value (NPV) for pulmonary artery acceleration time to right ventricular ejection time (PAAT:ET) measurements obtained at baseline, DOL 2, and DOL 5–7. ECMO, extracorporeal membrane oxygenation.

Cohort	n	Time of Echo	Outcome	AUC (95% CI)	*p*-Value	PAAT:ET Cut-Off	Patients below Cutoff (n, %)	Sensitivity	Specificity	PPV	NPV	Relative Risk
Entire cohort	87	Baseline	ECMO	0.815 (95% CI 0.717–0.913)	<0.001	≤0.290	41 (47.1%)	79.5%	79.2%	75.6%	82.6%	4.3
Entire cohort	87	Baseline	Mortality	0.715 (95% CI 0.569–0.860)	0.006	≤0.256	25 (28.7%)	70.6%	81.4%	48.0%	91.9%	6.0
	87	DOL 2	Mortality	0.745 (95% CI 0.603–0.886)	0.002	≤0.290	32 (36.8%)	82.4%	70.0%	40.0%	94.2%	6.9
	82	DOL 5–7	Mortality	0.866 (95% CI 0.749–0.984)	<0.001	≤0.303	22 (26.8%)	81.3%	86.4%	59.1%	95.0%	11.8
Only ECMO-patients	39	Baseline	Mortality	0.496 (95% CI 0.309–0.683)	0.966							
	39	DOL 2	Mortality	0.646 (95% CI 0.465–0.827)	0.123							
	34	DOL 5–7	Mortality	0.788 (95% CI 0.632–0.944)	0.004	≤0.310	19 (55.9%)	81.3%	66.7%	68.4%	80.0%	3.4

## Data Availability

The data presented in this study are available on request from the corresponding author. The data are not publicly available in accordance with the data protection law of Germany.

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
