# Peer review of "Echocardiographic Assessment of Pulmonary Hypertension in Neonates with Congenital Diaphragmatic Hernia Using Pulmonary Artery Flow Characteristics"

_jcm, 2022, doi:10.3390/jcm11113038_

Round 1

Reviewer 1 Report

I am honored to be one of the reviewers of this article. I believe that the use of PAAT:ET can be a very useful tool in the future. This study was conducted on a consistent group of cases, but the results may become statistically strong enough on a larger study. So, I advise you to continue enrolling other cases for next years. 

I would like to know what percentage of the cases were diagnosed antenatally. 

To be corrected:

Please describe the final study group more accurately and correct the manuscript the number of newborns (87 not 89).

The description of the study group must me very clear. You actually excluded 22 patients from final study, not 20. 

Author Response

Dear madame or sir, 

thank you very much for your review and the raised concerns that helped to improve our paper. To improve the quality of the manuscript we had a English native speaker check the manuscript, which led to some changes throughout the manuscript.

We continue indeed measuring PAAT:ET in all neonates with CDH (additionally in preterms) and we hope for international collaboration in a multi-center study.

In our study, 94.3% of patients were diagnosed antenatally, with a similar proportion in ECMO and non-ECMO patients. We included this information in table 1.

Additionally, we corrected the number of included patients in the revised manuscript. Thank you very much for mentioning this.

Reviewer 2 Report

This research was conducted to evaluate PH in CDH neonates. The study population is not small, and 39 patients required ECMO in five years. The results are interesting for the readers. However, I have several questions as below.

  1. Table 1:
  • Could the authors show the birthweight of participants?
  • I want to know when (gestational weeks) o/eLHR was measured.
  1. Figure 1: Please show the three groups by color.
  2. Figure 1: Could you evaluate the change of PAAT: ET between pre-and-post ECMO and compare them in ECMO-survivors and ECMO-non-survivors. That value seemed to increase in ‘ECMO- survivors but unchanged in ECMO-non-survivors.
  3. Discussion; Is the PAAT: ET at the baseline more effective in predicting the requirement of ECMO or the responsibility of ECMO?

Author Response

Dear madame or sir, 

thank you very much for your review and your comments. 

  1. We included information on birthweight and the timing of fetal ultrasound assessment for o/e LHR (26-28 weeks gestational age) in table 1 in the revised manuscript.
  2. Figure 1 is replaced by a figure using colors for the respective groups.
  3. This is an important point. We evaluated the relative changes and included this information in the results section: "The median PAAT:ET value at DOL 2 and DOL 5-7 was 101% and 103% of the baseline value in ECMO non-survivors and 128% and 139% in ECMO survivors, respectively. 35.3% and 31.3% of ECMO non-survivors and 54.5% and 72.2% of ECMO survivors presented an increase of more than 10% from baseline to DOL 2 and DOL 5-7, respectively."
  4. When comparing the positive and negative predictive values of PAAT:ET at baseline for predicting ECMO and at DOL 5-7 for predicting non-survival in ECMO patients, we observed similar values for predicting the requirement for ECMO and for a response to ECMO. We included this point in the discussion section of the revised manuscript.